# Responses of Soil Microbial and Nematode Communities to Various Cover Crop Patterns in a Tea Garden of China

**DOI:** 10.3390/ijerph19052695

**Published:** 2022-02-25

**Authors:** Lili Wang, Yang Wang, Weiming Xiu, Bingchang Tan, Gang Li, Jianning Zhao, Dianlin Yang, Guilong Zhang, Yanjun Zhang

**Affiliations:** 1Agro-Environmental Protection Institute, Ministry of Agriculture and Rural Affairs, Tianjin 300191, China; wy940215ok@163.com (Y.W.); xiuweiming@caas.cn (W.X.); tanbingchang@caas.cn (B.T.); ligang06@caas.cn (G.L.); zhaojianning@caas.cn (J.Z.); yangdianlin@caas.cn (D.Y.); zhangguilong@caas.cn (G.Z.); zhangyanjun@caas.cn (Y.Z.); 2Key Laboratory of Original Agro-environment Pollution Prevention and Control of Ministry of Agriculture and Rural Affairs, Tianjin Key Laboratory of Agro-Environment and Agro-Product Safety, Tianjin 300191, China

**Keywords:** cover crops, microorganisms, nematodes, soil food web, tea

## Abstract

As one of the typical farmland ecosystems, tea gardens are of vital importance in China. The purpose of this study was to quantify the dynamic of soil properties, soil microbial diversity, and nematodes, as affected by various cover crop patterns in a Tanjiawan tea garden in Hubei Province, China. Four cover crop patterns were established as following: naturally covered of bare land and mixed planting with two species, four species, and eight species. The results revealed that soil organic matter, pH, and total phosphorous content were significantly related to cover crop patterns. The number of nematodes increased with cover crop diversity, and the percentage of plant parasitic nematodes in cover crop treatments was lower than in naturally covered bare land. A higher diversity of cover crops increased the number of bacterivores and fungivores, thereby enhancing the bacterial decomposition pathway of soil organic matter. Both soil nematodes and microbial communities showed significant seasonal changes under different cover crop patterns. The soil food webs were more stable and mature under cover crops with two species and four species. Combined with the results of a structural equation model, we found that soil properties, characterized by the total nitrogen, available phosphorus, NO_3_^-^-N, and soil organic matter, were significantly correlated with soil nematodes and microbial communities. In addition, acterivores and plant parasites were significantly negatively correlated with omnivores/predators. Our results implied that soil properties and seasonal changes influence the relationships between cover crops, soil nematodes, and microbial communities. These findings provide a theoretical basis for future studies on interactions between soil properties, soil microorganisms, and nematodes in tea gardens.

## 1. Introduction

Biodiversity is an ecological complex formed by organisms and their surrounding environment, as well as the synthesis of various ecological processes [1]. Biodiversity is the basis for water conservation, soil conservation, climate regulation, and other ecosystem functions [2,3]. With agricultural intensive practices, multi-scale habitat heterogeneity has been significantly reduced, resulting in visible soil diversity degradation [4]. This will largely LIMIT the sustainable resources utilization related to soil diversity, thereby leading to the loss of natural resources which are substantial for human survival [5]. Understanding the response of biodiversity to different land use practices and climate change is important for preventing its decline [6,7]. Increasing the availability of non-cultivated habitats, i.e., cover crop, is an effective measure to protect farmland biodiversity [2,8]. Vegetation has the greatest impact on biodiversity and is vital for increasing species diversity [3]. When the diversity of plant species in an ecosystem changes, soil fertility, microclimate maintenance, and other ecosystem functions might also change, which is critically important for ecosystem function [5].

Previous studies have reported that planting cover crops has the potential to increase the richness of plant functional groups, thereby promoting soil microbial diversity, which can reduce the proliferation of soil-borne pathogens and increase the number of beneficial microorganisms [2,9]. Planting cover crops can also increase soil carbon turnover rate and nitrogen availability [10,11,12], thus providing favorable conditions for microbial proliferation [13]. In addition to act as nutrients, their root exudations are biodegradable organic substances which can be utilized as energy and nutrients for microbial growth, potentially influencing soil microbial diversity [14]. Additionally, different types of cover crops have different effects on soil microbial abundance [15]. For example, soil microbial biomass in the field of winter rapeseed used as cover crop is higher than in Violet and Hairy vetch fields [8]. Therefore, it is important to explore the dynamics and related mechanism of soil microbial diversity in various patterns of cover crops.

Soil nematodes are sensitive to their habitat changes [16]. Meanwhile, their structure and enrichment indices have been proven to be significantly correlated with soil carbon and nitrogen content [17], which can provide unique information on soil ecological processes [18]. The evaluation of soil nematodes community structure can be used as an index for soil decomposition pathway [19]. Therefore, soil nematodes can be regarded as a good indicator for the degree of soil disturbance and soil biodiversity [20]. Additionally, changes in soil nematode community structure are important for studying the status and function of soil organisms in vegetation succession [21]. It has been reported that the spatial distribution and total abundance of nematodes are greatly affected by cover crops, due to changes in soil carbon [22], as well as the increased structure and complexity of the soil food web [23]. However, although soil nematodes vary with different vegetation types [24], little research has been conducted on investigating plant varieties when choosing cover crops.

Jiang et al. [25] have demonstrated that there are interactions between soil microorganisms and nematodes. First, plant parasitic nematodes (PPNs) can change the root secretion morphology and chemical composition by infecting plants [26], provide more material energy and a better root chemical composition for soil microorganisms [27], as well as changing the community structure of rhizosphere fungi [26]. For example, root-parasitic nematodes can enhance soil microbial activity and promote soil microbial activities and nitrogen cycling [27]. Microbivorous nematodes could also affect soil microbial activities, influenced by plant species, substrate C/N ratios, as well as complex interactions between nematodes and other soil animals [28]. Due to the regulating effect of soil nematodes, the balance between beneficial parasitic microorganisms and pathogenic microorganisms in the soil is maintained [29], which is important for the prevention and treatment of harmful microorganisms, especially soil-borne pathogenic microorganisms [30]. Second, the antagonistic relationship between nematophagous microorganisms and PPNs provides an important reference for the biological control of nematodes. There are two ways in which microorganisms control nematode worms: (1) microorganisms trap and kill nematode worms from outside; and (2) microorganisms enter nematode worms parasitically and produce toxins to kill nematode worms [31]. Therefore, it is essential to explore how both soil nematodes and microorganisms could provide new biological control strategies for PPNs in the future [32].

Tea gardens, a typical agricultural ecosystem of China, have been influenced by the traditional concept of “grass and tea plants competing for fertilizer and water” in recent years. The research on cover crops started relatively recently, and few studies have been conducted in tea gardens. We hypothesized the following: (1) diversified cover crop patterns exert different effects on soil microorganism and nematode communities in a tea plantation; (2) soil physical and chemical factors are interacted with dynamics of soil nematodes and microbes, as affected by planting diversified cover crops; (3) soil microorganism and nematode communities show different changing trends under the diversified cover crop patterns, depending on seasonal changes. The findings of this study will reveal the effect of various cover crop patterns on soil fertility and soil food webs in a tea garden.

## 2. Materials and Methods

### 2.1. Study Site

The experimental site was located at Weiping Temple Village, Tanjiawan Town, Yunyang District, Shiyan City, Hubei Province, China (32°93′ N, 110°87′ E), which has an altitude of 220 m, an annual rainfall of 800–1100 mm, a frost-free period of 248 d, a north sub-tropical continental monsoon climate, with annual average sunshine hours of 1655–1958, and an annual average temperature of 16 °C. According to the United Nations Food Agriculture Organization (FAO), soil type is mainly Humic Cambisols. Soil pH was around 6.02–6.48 during the study period.

### 2.2. Experimental Design

The experiment started from 2016, and there were four cover crop patterns with three replicates: naturally covered bare land (A0), mixed planting with two species (*Lolium perenne* and *Trifolium repens*) (A1), mixed planting with four species (*L. perenne*, *T. repens*, *Poa annua*, and *T. pratense*) (A2), and mixed planting with eight species (*L. perenne*, *T. repens*, *P. annua*, *T. pratense*, *Festuca rubra*, *Vicia villosa Roth*, *Perennial Coreopsis*, and *Zinnia elegans Jacq*) (A3). During the experiment, weeds were regularly removed between rows. All treatments were treated with conventional field management measures. The area of each treatment was 1200 m^2^, with 400 m^2^ for each replicate. The spacing of tea planting rows was 1.5 m. The sowing weight of covered crops is shown in Appendix A.

### 2.3. Soil Samples

Soil samples were collected in May, August, and November 2018, as well as February 2019. Fifteen points were selected for each soil sampling. Soil samples (0–20 cm) were taken by drill with a diameter of 3 cm. Five triplicates were selected for each treatment, and soil samples were divided into three parts. The first part was dried at room temperature and then used for the analysis of soil physical and chemical properties. The second part was stored at −70 °C for soil microbial analyses, and the third part was stored at 4 °C for the isolation and subsequent identification of soil nematodes.

### 2.4. Analysis of Soil Physical and Chemical Properties

The soil nitrate (NO_3_^−^-N) and ammonium (NH_4_^+^-N) were extracted using the CaCl_2_ extraction method and measured by an AA3 flow analyzer (Bran Lubbe AA3, Hamburg, Germany). The soil total nitrogen (TN) was determined by a flow analyzer (Bran Lubbe AA3, Hamburg, Germany) [33]. The Soil pH was determined with the glass electrode method using an MP511 pH meter (MP511 pH meter, Shanghai, China) [34]. The ratio of soil to water was 1:2.5. The soil organic matter (OM) was determined by a total organic carbon analyzer (Multi N/C3100, Hamburg, Germany) [33]. The soil available phosphorus (AP) and the soil total phosphorus (TP) were determined by methods described by Lu [35]. The AP was extracted using 0.5 mol/L NaHCO_3_ and determined by the molybdenum–antimony anti-spectrophotometric method (UV-2800 Ultraviolet-visible Spectrophotometer, Shanghai, China). The TP was boiled with HClO_4_–H_2_SO_4_ and determined using the molybdenum–antimony anti-spectrophotometric method (UV-2800 Ultraviolet-visible Spectrophotometer, Shanghai, China). The microbial biomass carbon (MBC) and the microbial biomass nitrogen (MBN) were fumigated by chloroform. Ultimately, 20 g of fresh soil were fumigated by chloroform at 25 °C for 48 h without light exposure [36]. Their contents were extracted by potassium sulfate and determined by a total organic carbon analyzer (Multi N/C3100, Hamburg, Germany). The conversion factor (0.45 for the MBC and 0.54 for the MBN) was calculated according to the difference between the fumigated and non-fumigated subsamples [37].

### 2.5. Nematode Identification

A modified shallow tray method was utilized for separating nematodes from soil [38]. Then, 50 g of fresh soil were weighed and evenly spread on a filter paper in a tray. After a 48 h separation at 25 °C, soil leaching liquid was filtered through a 500 μm mesh sieve. The rinse solution was collected in a sample bottle and was left to stand for two hours. The rinse solution was carefully extracted, and 9 mL were retained. Nematodes were then concentrated in water at the bottom of the sample bottle. The sample bottle must not be shaken during operation to prevent nematodes from being agitated. Nematodes were killed through heating in a water bath at 60 °C for 3 min and then cooled by standing. One milliliter of a nematode fixing solution (80% formalin, 20% glycerol, and two to three drops of glacial acetic acid) was added. Nematodes were counted with an aspheric microscope, and the number of nematodes per 100 g dry soil was calculated. Nematode worms were identified under a high-power (200×) inverted fluorescence microscope. Morphological methods were used to identify the taxonomic units at the genus level [39].

According to the identification results, the nematodes could be divided into different c-p groups and four functional nutrition groups: bacterivores (Ba), plant parasites (Pp), fungivores (Fu), and omnivores/predators (Op) [40,41].

Based on the classification results, the ecological and structural indices of nematodes were calculated as follows [41,42,43,44,45,46]:

The Shannon–Wiener diversity index (H) was calculated as following:H = −∑*Pi* ln *Pi* (*i* = 1, 2, 3,…, S),
where *Pi* is the ratio of the individual number to the nematode quantity in taxon i of the sample, and S is the number of the identified taxa.

The Pielon’s evenness index (J) was calculated as following:J = H/lnS.

The Simpson dominance index (λ) was calculated as following:λ = ∑*pi*^2^ (i = 1, 2, 3, …, S).

The structure index (SI) was calculated as following:SI = 100 × (s/(s + b)),
where s represents nematodes with c-p values of 3, 4, or 5 in the Ba, Fu, and Op groups, respectively, and b is the nematodes with a c-p value of 2 in the Ba group.

The enrichment index (EI) was calculated as following:EI = 100 × (e/(e + b)),
where e represents nematodes with a c-p value of 1 in the Ba and Fu groups.

The nematode pathway index (NCR) was calculated as following:NCR = N_B_/(N_B_ + N_F_),
where N_B_ is the number of nematodes in the Ba group, and N_F_ is the number of nematodes in the Fu group.

The Wasilewska index (WI) was calculated as following:WI = (N_F_ + N_B_)/N_PP_,
where N_PP_ is the number of nematodes in the Pp group.

### 2.6. Microbes

The soil microbial community was characterized by the phospholipid fatty acid (PLFA) method [47]. Three grams of freeze-dried soil were accurately weighed, and PLFAs were extracted by adding a mixed agent (chloroform:methanol:citrate volume ratio: 1:2:0.8). After a full oscillation on a horizontal oscillator, the supernatant was collected in a centrifuge. Citric acid buffer (the volume ratio of the citric acid solution:trisodium citrate solution: 5.9:4.1) and chloroform were added and then left overnight in the dark. Phospholipids were separated from neutral lipids and glycolipids by an SPE column, collected in methanol and dried with nitrogen. The phospholipids were converted into PLFA methyl ester using mild and basic methyl ester and then added to an internal standard solution (nonadecanoic acid and methyl ester). The types and contents of PLFAs were detected by GC-MS (6890-5973N). The corresponding soil microbial communities were characterized by the structural diversity and biological specificity of phospholipid fatty acids and classified into the following: gram-positive bacteria (G+), gram-negative bacteria (G−), bacteria (B), fungi (F), and actinomycetes (A) (Appendix A) [48].

The ecological index of each community was calculated as follows [49,50,51].

The Shannon–Wiener diversity index (H) was calculated as following:H = −∑*Pi*㏑*Pi* (I = 1, 2, 3, …, S),
where *Pi* is the ratio of the number of PLFAs to the total PLFAs in taxon i of the sample, and S is the number of microbial taxa.

The Pielon’s evenness index (J) was calculated as following:J = H/lnS.

The Simpson dominance index (λ) was calculated as following:λ = ∑*pi*^2^ (*i* = 1, 2, 3, …, S).

The Margalef richness index (D) was calculated as following:D = (S − 1)/lnN,
where N is the total number of PLFAs.

### 2.7. Statistical Analyses

SPSS Statistics v21.0 was used to conduct one-way and two-way ANOVA analyses on the test data. Duncan’s new multiple range test and Fisher’s least significant difference (LSD) post hoc tests were used for multiple comparisons in order to determine significant differences between treatments with a significance level of *p* < 0.05. Origin v9.4 was used for creating figures, and Amos v21.0 was used for a structural equation model (SEM) construction [52]. A redundancy analysis (RDA) was conducted using Canoco v5.0.

## 3. Results

### 3.1. Soil Physical and Chemical Properties

The soil physical and chemical properties under different cover crop patterns showed variable trends, depending on the specific indices. The OM, pH, and TP were significantly related to cover crop patterns (*p* < 0.05; Appendix A). Among the different treatments, TP content was higher in A1 and A3, compared to in A0, even though there was no statistical difference in the TP contents between A0 and A2. Compared to in A0, pH was higher in the covering planting modes in A1, A2, and A3, with the highest value occurring in A2 (*p* < 0.05; Table 1).

Our results showed that the water content of soil (WCS), pH, TP, NO_3_^−^, NH_4_^+^, AP, and OM were also significantly related to months (*p* < 0.05; Appendix A), implying the importance of seasonal changes’ effects on soil physical and chemical properties in a tea garden.

### 3.2. Soil Nematode Communities

The number of nematodes increased with cover crop diversity. The number of nematodes in A3 was the highest in May and November 2018 and February 2019. There were no significant differences in the nematode numbers between A1, A2, and A3 in August 2018, which were all higher than that in A0 (Figure 1). As shown in Figure 2, the percentages of PPNs in treatments with cover crops in A1, A2, and A3 were lower than in A0 (*p* < 0.05). There were no significant differences in the percentage of omnivores/predators and PPNs under different cover crop patterns. Our results showed that increased cover crop diversity reduced the number of PPNs (Figure 2).

The Shannon–Wiener diversity indices of soil nematodes were not significantly affected by the increased cover crop diversity. Compared to in A0, the Simpson dominance indices of cover treatments in A1, A2, and A3 were significantly lower (*p* < 0.05). The results revealed that cover crops increased the uniform distribution of species. The Wasilewska indices in treatments with cover crops were significantly higher in A2 and A3 (*p* < 0.05) were higher than in A1. Moreover, the Wasilewska indices showed no significant difference between A2 and A3. The diversity of cover crops did not affect the decomposition pathway of soil OM (*p* > 0.05). The nematode channel ratio (NCR) of >0.5 indicated that the bacterial channel served as the main pathway of soil OM decomposition in the four treatments (Table 2). However, the NCRs in plots with cover crops in A1, A2, and A3 were significantly higher than in A0 (*p* < 0.05), indicating that cover crop enhanced the bacterial decomposition pathway of soil OM.

As shown in Figure 3, compared to in A0, the flora changes in A1 and A2 were more concentrated in quadrant B, indicating that A1 and A2 were more conducive. It implied that soil environment risk was reduced and a more stable and mature food network was enabled.

### 3.3. Soil Microbial Community

In May 2018, the total number of PLFAs in microorganisms in A0 was the largest, and there were no significant differences in the total number of PLFAs among A1, A2, and A3 (Figure 4). In the other three months, no significant differences were found among the four cover crop patterns.

As shown in Figure 5, the interaction of the monthly variation and cover crop patterns showed no significant effect on the soil microbial community. The percentage of soil fungi and common bacteria in different months under various cover crop patterns was shown in Appendix A. Our results found that the number of soil microorganisms was mainly affected by seasonal changes, and cover crop diversity only had a significant impact on the number of soil fungi (Appendix A). Moreover, there were no significant differences between soil microbial indices, as cover crop diversity increased (Appendix A). 

Ecological indices of the microbial community were shown in Table 3, but there are no significant differences among the various cover crop patterns. The RDA results showed that the NH_4_^+^-N, TN, and TP were positively correlated with the Pielou’s evenness, Shannon–Wiener diversity, and Margalef richness indices for microorganisms but negatively correlated with the Simpson dominance index. The TP content was significantly positively correlated with the Margalef richness index, indicating that soil microbial population richness rose with the increased TP content. The WCS, OM, and pH were negatively correlated with soil microbial Pielou’s evenness, Shannon–Wiener diversity, and Margalef richness indices but positively correlated with the Simpson dominance index. The AP was positively correlated with the Pielou’s evenness and Shannon–Wiener diversity indices of soil microorganisms but negatively correlated with the Margalef richness index, indicating that the AP content promoted the enhancement of biodiversity and evenness of soil microbial communities but inhibited their species richness (Figure 6).

### 3.4. Mechanism of the Potential Impact of Cover Crops on Soil Food Webs

As shown by SEM (Figure 7), soil properties, characterized by the WCS, TN, AP, NO_3_^−^-N, and OM, were significantly correlated with the community structure of microorganisms and nematodes (*p* < 0.05). There was a significant correlation between soil physical and chemical factors and the community structure of microorganisms and nematodes (*p* < 0.05). The correlation coefficient between soil physical and chemical factors and bacteria was −0.52, and its correlation coefficient with fungi was 0.62. The relationship between soil properties and bacteria or fungi was stronger than with PPNs. In our study, soil nematodes showed no significant correlation with microorganisms. Fungivores, bacterivores, and plant parasites were significantly negatively correlated with omnivores/predators, among which the most significant correlation was between plant parasites and omnivores/predators (*p* < 0.05; Figure 7).

## 4. Discussion

### 4.1. Effects of Cover Crop Patterns on Soil Nematode Community Characteristics and the Related Ecological Index

It has been reported that, with exception of PPNs, the abundance of other vegetative groups increased with a higher cover crop diversity [22,53]. In our study, nematode density was not significantly different among A1, A2, and A3 (Figure 2 and Appendix A), but the nematode densities in these three groups were higher than in A0. The percentage changes in PPNs and the patterns of nematode density exhibited opposite trends (Figure 2), indicating that a higher cover crop diversity mainly increased the number of bacterivores and fungivores. This might be due to that biodiversity of cover crops increased the diversity of plant litters and rhizospheric sediments. It has already been verified that the decomposition of different litters and rhizospheric sediments promotes the accumulation of soil humic acid, which affects the proportion of bacterivores and fungivores [54]. Different plant functional groups can influence the composition of soil nematode communities. In a previous study, it was found that leguminous plants could increase the number of bacterivores, while the number of fungivores was positively correlated with the number of weeds [55]. Bachie and McGiffen [56] reported that cover crops inhibit the growth of weeds and indirectly lead to a decrease of fungivores, consistent with our results of soil nematodes during winter in this study.

Meanwhile, it has been reported that the succession of PPNs is more easily affected by changes of the plant community than the other way round [57]. As for the nematode community composition, plant species identification is often more important than plant diversity [55]. A combination of Perennial Coreopsis and Z. elegans Jacq inhibits the reproduction of parasitic nematodes and thereby affects their densities [58,59]. It is also reported that, unlike leguminous plants, gramineous plants inhibit the growth of nematode density [60,61]. However, cover crops belonged to the mixed sowing of gramineae and leguminous plants in our study, and no significant differences in PPNs were found between different treatments. These results indicated that there might be interactions between leguminous and gramineous plants and soil nematodes, which remains to be elucidated.

Previous studies demonstrated that the decomposition of OM depends on the abundance of easily decomposed parts of OM [62]. When the OM was abundant and easy to decompose, the decomposition of the OM in soil food web is mainly through the bacterial channel. In contrast, when the OM is poor and difficult to decompose, the decomposition of the OM in the soil food web is mainly through the fungi channel [62,63]. In this study, the NCR was >0.5, which indicated that the bacterial channel acted as the main pathway of the OM decomposition. Additionally, plots covered with crops in A1, A2, and A3 had a significantly higher OM decomposition than in A0. These results showed that cover crops increased the bacterial decomposition pathway of the OM. Changes in the soil nematode community structure caused by environmental factors are always closely related to the ecological index of nematodes [64]. The findings of this study clearly illustrated the relationships among soil nematodes community composition, environmental changes, and ecosystem functions of nematodes, as affected by cover crops in tea gardens.

### 4.2. Effects of Cover Crop Patterns on Microbial Community Characteristics and the Related Ecological Indices

Finney et al. [65] verified that species-specific cover crops have different effects on the soil microbial community composition. In our study, the increased diversity of cover crops did not significantly influence the total PLFAs of microorganisms, but fungi varied among different cover crop patterns. This could be due to the divergent microbial catabolic activity and the concentrations of aromatic organic compounds produced by different cover crop species [66], and the effects of cover crops on soil bacteria might be neutralized by different cover crop species. Overall, the number of soil microbial communities in our study was mainly affected by seasonal changes, which indicated that the proportion of each microbial group (except for actinomycetes) did not change significantly under different cover crop patterns (Figure 5; Appendix A). The proportions of actinomycetes in the cover crop patterns in A1, A2, and A3 were lower than those in A0, and significant differences were found between A1, A2, and A3. This might be related to differences in the soil environmental preference for actinomycetes in the various cover crop patterns. Meanwhile, it has been shown that actinomycetes are the most significant factor affecting the TP [67]. In our study, except A2, the TP contents of cover crops in mixed plots were higher than that in A0, which was consistent with a previous study regarding the effect of cover crops on the soil surface phosphorus [68], which reported that fertilizer P could be reduced in such cover crop systems.

Soil important functions as well as the degree of soil health are always affected, combined with changes in the soil microbial community composition and soil physical-chemical properties responsive to cover crops [69,70]. In our study, the RDA results on the ecological index of soil microorganisms and soil physical-chemical properties revealed that the TP content was correlated with the richness of soil microbial population. In addition, the AP content had a positive relationship with the biodiversity and evenness of soil microbial community, while it had a negative relationship with soil microbial species richness. Our findings verified that soil properties and seasonal changes deserved increased attention when exploring the effects of cover crops on the ecological indices of soil microbial communities. This study is critical in improving the ability to better predict potential changes in the soil function and soil health, as affected by cover crops.

### 4.3. Analysis of the Soil Food Web

SEM could help uncover the direct and indirect relationships in complex soil food webs [71]. The enrichment index reflects the main nutritional level of the food network, which could represent the richness and activity of main detrital consumers [41]. A flora analysis on soil nematodes combines functional and reactive factors, which has been proven helpful in a previous study on the mutual relationship between the soil nematode diversity and the ecosystem function [72]. In our study, soil properties had an influence on soil nematodes and microorganisms, among which the WCS exerted the greatest effect. Soil physical and chemical properties were negatively correlated with the proportion of soil bacteria but positively correlated with the proportion of soil fungi. Changes in soil bacteria and fungi reflected the degree of soil health and thereby affected the food web structure. Soil properties mainly affected plant parasites and showed a negative correlation. The ratio of omnivores/predators negatively correlated with bacterivores, fungivores, and plant parasites, mainly due to the predatory relationships between these organisms [40,53]. In this study, the correlation between omnivores/predators and plant parasites was the strongest, indicating that the omnivores/predators in soil food web mainly feed on plant parasites. The diversity of cover crops influenced PPNs by affecting soil properties as well as the community structure of soil nematodes through the energy flow in the food web.

## 5. Conclusions

In this study, we found that increasing the diversity of cover crops enhanced the total density of nematodes, which thereby increased the number of bacteria and fungi-eating nematodes, but inhibited the number of PPNs. Both soil nematodes and microbial communities showed significant seasonal changes. The soil food web was found to be more stable and mature under two and four crop mulching patterns. The diversity of cover crops affected PPNs by changing soil physical and chemical properties and the community structure of nematodes via energy flows in the food webs. In addition, significant relationships were found between soil microbial communities and soil properties, such as soil OM, TP, and pH. Our results demonstrated that the effect of cover crop on soil nematodes and microbial communities depends on soil properties and seasonal changes in tea gardens. This study highlights the significant interactions among soil properties, nematodes, and microbial communities, which implies the necessity of simultaneous studies on multiple factors when exploring soil function changes as influenced by diversified cover crop patterns in tea gardens.

Further studies should be designed on a multi-year basis, and long-term investigations of the automatic detection of soil temperature and humidity should be considered in studies on soil biodiversity, as affected by various cover crop patterns. In addition, both beneficial and harmful nematode and microorganism species should be analyzed, and these interactions should be studied in more detail in future studies.

## Figures and Tables

**Figure 1 ijerph-19-02695-f001:**
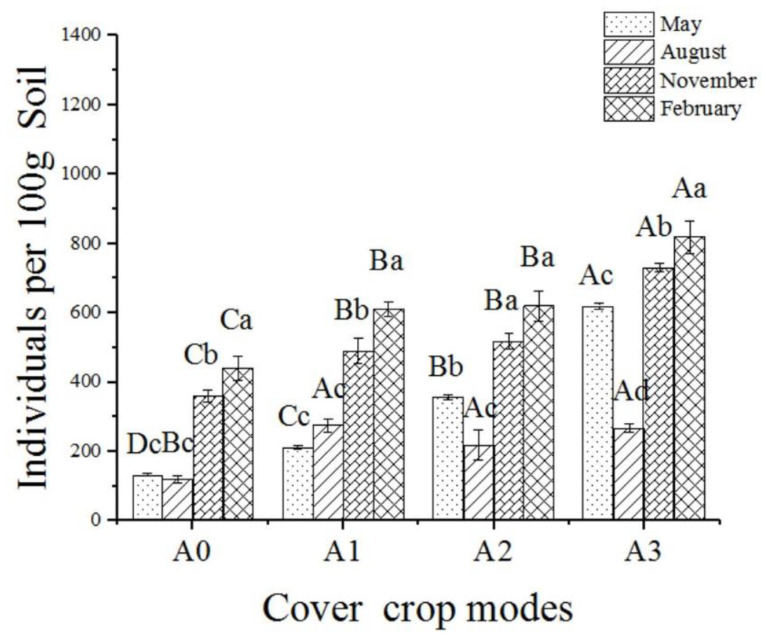
Variation of nematode abundance under different cover crop patterns. Note: A0 represents naturally covered of bare land; A1 represents mixed planting with two species; A2 represents mixed planting with four species, and A3 represents mixed planting with eight species. Different capital letters indicate the significant differences between different cover crop patterns in the same month, and different lowercase letters indicate the significant differences between different months of the same cover crop patterns (*p* < 0.05).

**Figure 2 ijerph-19-02695-f002:**
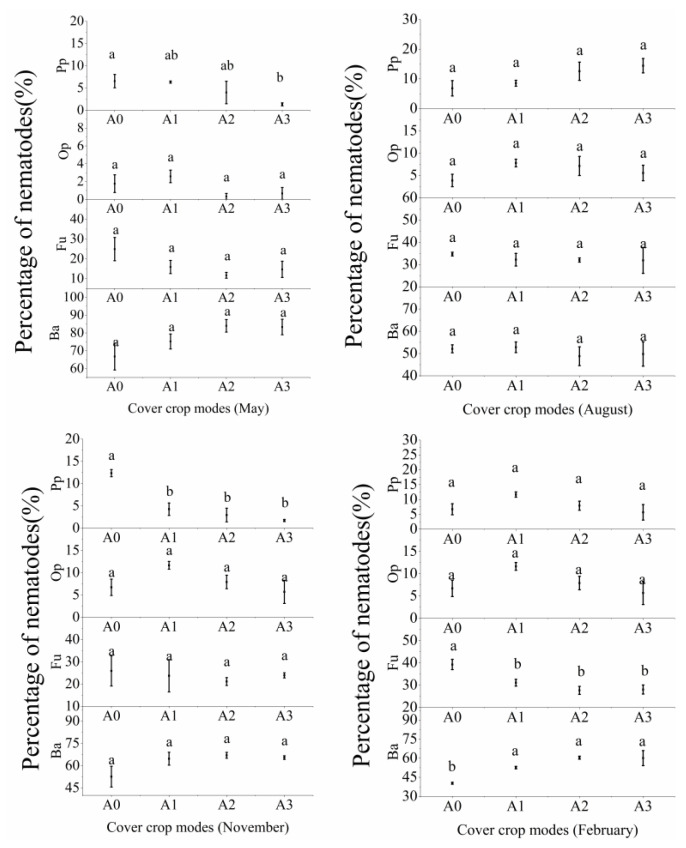
Changes in the nutritional groups of nematodes under different cover crop patterns. Different lowercase letters indicate that there are significant differences between different cover crop patterns in the same month (*p* < 0.05). Ba, bacterivores; Fu, fungivores; Op, omnivores/predators; Pp, plant parasites.

**Figure 3 ijerph-19-02695-f003:**
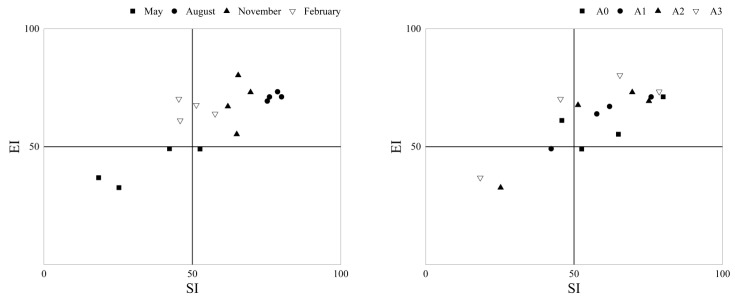
Flora analysis of nematode worms. EI, enrichment index; SI, structure index.

**Figure 4 ijerph-19-02695-f004:**
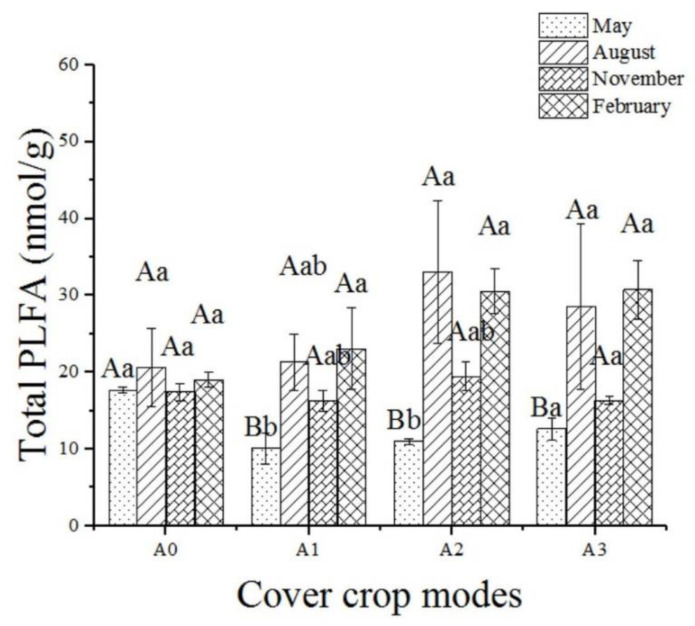
Variation of the total phospholipid fatty acids (PLFAs) in microorganisms. Different capital letters indicate the significant differences between different cover crop patterns in the same month, and different lowercase letters indicate the significant differences between different months in the same cover crop patterns (*p* < 0.05).

**Figure 5 ijerph-19-02695-f005:**
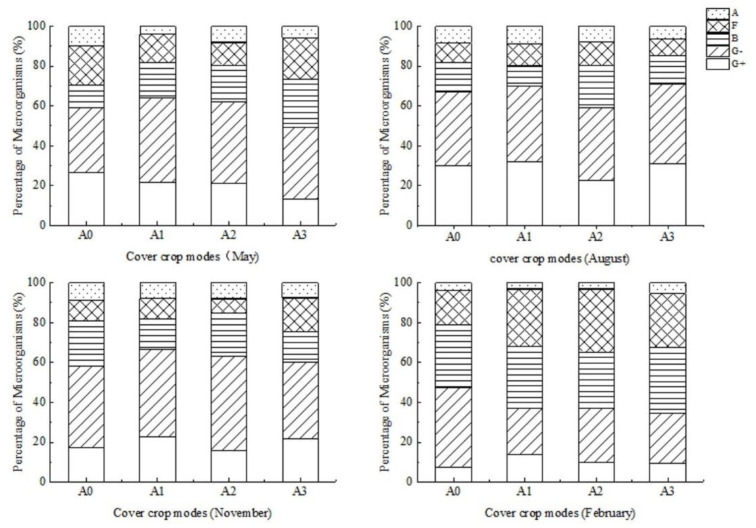
Changes of microbial communities under different cover crop patterns. G+, gram-positive bacteria; G–, gram-negative bacteria; A, actinomycetes; F, fungi; B, bacterial.

**Figure 6 ijerph-19-02695-f006:**
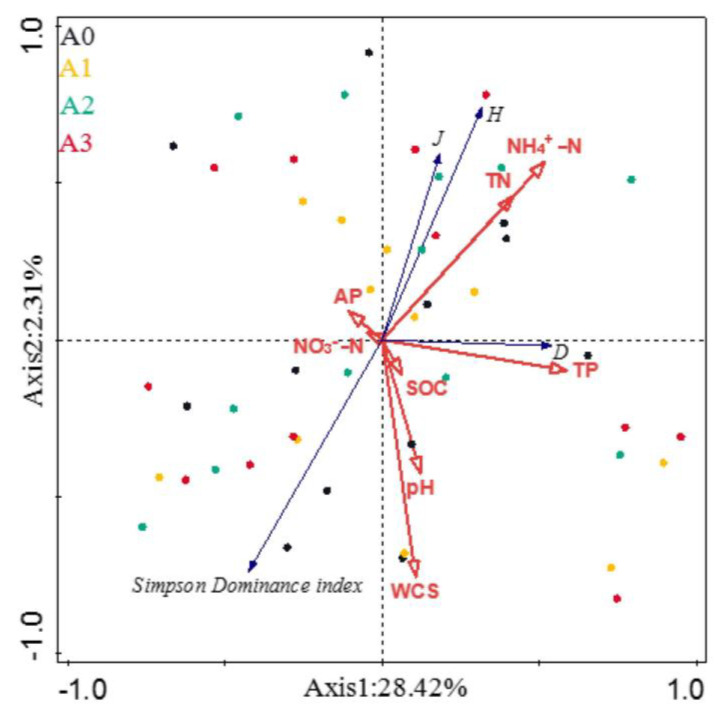
Correlation between the edaphic physicochemical factors and the microbial communities diversity index. J, Pielou’s evenness index; H, Shannon–Wiener diversity index; D, Margalef richness index; TN, total nitrogen; TP, total phosphorus; AP, available phosphorus; SOC, soil organic carbon; WCS, water content of soil; NO_3_^−^-N, nitrate; NH_4_^+^-N, ammonium.

**Figure 7 ijerph-19-02695-f007:**
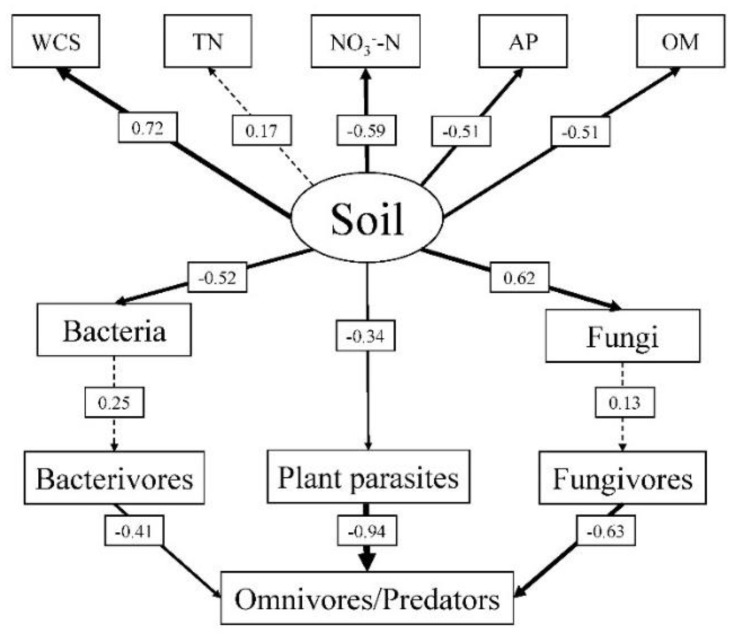
Feeding relationships in the soil food network under different cover crop patterns uncovered by SEM (X^2^ = 43.595, d.f. = 36, *p* = 0.180, GFI = 0.831, CFI = 0.874, RMSEA = 0.067). Solid lines represent significant correlations (*p* < 0.05) and dotted lines indicate nonsignificant correlations. WCS, water content of soil; TN, total nitrogen; AP, available phosphorus; OM, organic matter; NO_3_^−^-N, nitrate.

**Table 1 ijerph-19-02695-t001:** Soil physical and chemical properties under different cover crop patterns. Different lowercase letters indicate significant differences between different cover crop patterns (*p* < 0.05).

Cover Crop Pattern	A0	A1	A2	A3
water content of soil (WCS; %)	13.31 ± 1.00 a	13.35 ± 0.98 a	13.69 ± 0.99 a	13.17 ± 1.08 a
organic matter (OM g/kg)	1.34 ± 0.08 a	1.42 ± 0.06 a	1.52 ± 0.09 a	1.48 ± 0.08 a
pH	6.22 ± 0.05 c	6.43 ± 0.03 b	6.56 ± 0.03 a	6.35 ± 0.04 b
available phosphorus (AP; mg/kg)	12.67 ± 2.56 a	15.01 ± 1.99 a	10.69 ± 1.57 a	12.56 ± 2.20 a
total phosphorus (TP; g/kg)	0.43 ± 0.03 b	0.64 ± 0.03 a	0.45 ± 0.02 b	0.66 ± 0.04 a
total nitrogen (TN; g/kg)	0.65 ± 0.05 a	0.72 ± 0.04 a	0.76 ± 0.03 a	0.72 ± 0.04 a
nitrate (NO_3_^−^-N; mg/kg)	2.47 ± 0.31 a	3.32 ± 0.48 a	2.60 ± 0.41 a	2.85 ± 0.37 a
ammonium (NH_4_^+^-N; mg/kg)	4.20 ± 0.60 a	4.89 ± 0.71 a	3.77 ± 0.60 a	5.24 ± 0.62 a
microbial biomass carbon (MBC; mg/kg)	124.81 ± 16.51 a	134.78 ± 14.25 a	144.38 ± 16.02 a	153.06 ± 11.65 a
microbial biomass nitrogen (MBN; mg/kg)	17.44 ± 2.20 a	20.19 ± 1.87 a	20.75 ± 1.99 a	21.23 ± 2.38 a

Note: A0 represents naturally covered of bare land; A1 represents mixed planting with two species; A2 represents mixed planting with four species, and A3 represents mixed planting with eight species. The same representations were used throughout the paper.

**Table 2 ijerph-19-02695-t002:** Ecological indices of the nematode communities. Different lowercase letters indicate the significant differences between different cover crop patterns (*p* < 0.05).

Cover Crop Pattern	A0	A1	A2	A3
Shannon–Wiener diversity index (H)	2.236 a	2.253 a	2.248 a	2.245 a
Simpson dominance index (λ)	0.147 a	0.104 c	0.116 bc	0.119 b
Pielou’s evenness index (J)	0.853 a	0.842 a	0.853 a	0.847 a
Structure index (SI)	60.873 a	59.471 a	55.369 a	52.007 a
Enrichment index (EI)	59.164 a	62.835 a	60.700 a	65.192 a
Wasilewska index (WI)	12.845 b	18.778 ab	39.996 a	41.189 a
Nematode channel ratio (NCR)	0.626 b	0.703 ab	0.731 a	0.718 ab

**Table 3 ijerph-19-02695-t003:** Ecological indices of the microbial community. Different lowercase letters indicate the significant differences between different cover crop patterns (*p* < 0.05).

Cover Crop Patterns	A0	A1	A2	A3
Shannon–Wiener diversity index (H)	2.622 a	2.580 a	2.662 a	2.636 a
Simpson dominance index (λ)	0.090 a	0.094 a	0.088 a	0.089 a
Pielou’s evenness index (J)	0.980 a	0.980 a	0.990 a	0.985 a
Margalef richness index (D)	4.681 a	4.735 a	4.632 a	4.609 a

## Data Availability

Not applicable.

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
