# Peer review of "Responses of Soil Microbial and Nematode Communities to Various Cover Crop Patterns in a Tea Garden of China"

_ijerph, 2022, doi:10.3390/ijerph19052695_

Round 1

Reviewer 1 Report

This study aims to quantify the dynamic of soil properties, soil microbial diversity and nematodes as affected by various cover crop patterns in the Tanjiawan tea garden in Hubei Province, China. The research work topic is important and worth of investigation and approving. I accept the publication of this manuscript in its current form.

Reviewer 2 Report

The authors have improved the manuscript and have responded appropriately to my concerns. The manuscript is clearly written. The following issues should be fixed.

  1. The keyword Teashould be revised as Tea garden.
  2. Line 307, bacterial should be revised as bacteria.
  3. Line 323, the full name of WCS should be given.

Reviewer 3 Report

After re-reviewing the article, I find that all my comments on the first review have been taken into account. The article is interesting and can be published in the International Journal of Environmental Research and Public Health.

This manuscript is a resubmission of an earlier submission. The following is a list of the peer review reports and author responses from that submission.

Round 1

Reviewer 1 Report

Overall, the introduction, material and methods, results, and discussion, as well as conclusions are properly presented, confirming the high scientific expertise of the research team. This paper investigates the dynamics of microorganism and nematode communities, to uncover the relationship between soil physical and chemical factors as well as nematodes and microbes, as affected by planting diversified cover crops in a tea plantation. The research work topic is important and worth investigation and approving, however, there are many shortcomings that must be rectified. In general, important information is presented.

  1. L13; Please give more details “effects of different cover crop”
  2. Please subdivide this keyword “Physical and chemical soil properties”
  3. Please remove the abbreviation in the abstract

4.      This works has a variety of data which are not apparent by just reading the abstract. There appears to be some information, which can add to knowledge in this growing field.

  1. Some keywords did not match with the abstract; Please add Tea as a keyword
  2. L84-L87: please rephrase this sentence
  3. L89: Please add a reference
  4. L131-L147: authors must give a reference for each experience, or give more details about the experience.
  5. Please delete ‘.’ And ‘,’ after each equation
  6. In the discussion, author would have benefited from a better understanding of the existing literature.
  7. In some sentence, English appears not to be adequate.
  8. To use a space between the number and the unit, as 20 °C; and not to use a space between number and percentage, as 10%, for example.
  9. Conclusion: please include limitations and future research area!
  10. References must be revised.

Author Response

Dear Reviewer#1,

We are grateful to you for giving us this valuable opportunity to revise our manuscript entitled “Responses of microorganism and nematode communities to various cover crop patterns in a tea garden of China” (ID: ijerph-1467238).

Those comments have been very valuable and useful for revising and improving our manuscript, as well as help us guiding our future research. All the comments are carefully revised. We believe that the entire manuscript has been improved a lot after extensive revisions. Again, on behalf of all co-authors, we really appreciate your patience and enthusiastic participation on our manuscript.

Detailed responses are as follows:

  1. L13; Please give more details “effects of different cover crop”

Response: Thanks. We reorganized this sentence, as follows “The purpose of this study was to quantify the dynamic of soil properties, soil microbial diversity and nematodes as affected by various cover crop patterns in the Tanjiawan tea garden in Hubei Province, China.”

  1. Please subdivide this keyword “Physical and chemical soil properties”

Response: Thank you very much for your careful comments. We deleted this keyword, which is also suggested by the other reviewer.

  1. Please remove the abbreviation in the abstract

Response: Thanks. The suggestion has been implemented.

This works has a variety of data which are not apparent by just reading the abstract. There appears to be some information, which can add to knowledge in this growing field.

Response: Thank you very much for your valuable suggestions. We readjusted the abstract, and added more information about the results verified in this study. e.g. Soil organic matter, pH, and total phosphorous contents were significantly related to cover crop patterns.

The number of nematodes increased with cover crop diversity, and the percentage of plant parasitic nematodes in cover crop treatments was lower than naturally covered of bare land.

we found that soil properties, characterized by total nitrogen, available phosphorus, NO3--N and soil organic matter, were significantly correlated with soil nematodes and microbial communities. In addition, acterivores and plant parasites were significantly negative correlated with omnivores/predators.

  1. Some keywords did not match with the abstract; Please add Tea as a keyword

Response: Thanks. The suggestion has been implemented. We reorganize the Keywords.

The original Keywords (Cover crops; Physical and chemical soil properties; Microorganisms; Nematodes; Structural equation models) was replaced by the following one:

Keywords: Cover crops; Microorganisms; Nematodes; Soil food web; Tea.

  1. L84-L87: please rephrase this sentence

Response: Thanks. We feel very sorry for the confusing expression in the previous version. We reread the related references, and the sentence has been reorganized as follows: For example, root-parasitic nematodes can enhance soil microbial activity, promote soil microbial activities and nitrogen cycling (Tu et al. 2003). Microbivorous nematodes could also affect soil microbial activities, influenced by plant species, substrate C/N ratio, as well as complex interactions between nematodes and other soil animals (Wu et al. 2007).

  1. L89: Please add a reference

Response: The suggestion has been implemented.

Cumagun, C. J. R., Moosavi, M. R. Significance of biocontrol agents of phytonematodes. Askary, T. H.;Martinelli, P. R. P(Editor). Biocontrol agents of phytonematodes.2015. CABI,Wallingford. Page 50-78. DOI:10.1079/9781780643755.0050.

  1. L131-L147: authors must give a reference for each experience, or give more details about the experience.

Response: Thank you very much for your careful comments. We added the related references. Also, we give more details. For example, the conversion factor (0.45 for MBC, 0.54 for MBN) was calculated according to the difference between fumigated and non-fumigated subsamples (Joergensen and Müller, 1996).

Wu, J., Joergensen, R.G., Pommerening, B., Chaussod, R., Brookes, P.C., 1990. Measurement of soil microbial biomass C by fumigation-extraction-an automated procedure. Soil Biol. Biochem. 22, 11671169.

Song, L., Tian, P., Zhang, J.B., Jin, G.Z. 2017. Effects of three years of simulated nitrogen deposition on soil nitrogen dynamics and greenhouse gas emissions in a Korean pine plantation of northeast China. Science of the Total Environment, 609, 1303-1311

Rhoades, J., 1996. Salinity: electrical conductivity and total dissolved solids. In:Sparks, D.L. (Ed.), Methods of Soil Analysis, Part 3. Chemical Methods.

Joergensen and Müller, 1996. The fumigation-extraction method to estimate soil microbial biomass: calibration of the kEN value. Soil Biol. Biochem., 28 (1996), pp. 33-37, 10.1016/0038-0717(95)00101-8.

Lu, R.K. 1999. Analysis Methods of Soil Science and Agricultural Chemistry. Agriculture Science and Technology Press, Beijing.

Please delete ‘.’ And ‘,’ after each equation

Response: The suggestion has been implemented.

  1. In the discussion, author would have benefited from a better understanding of the existing literature.

Response: Thank you very much for this valuable suggestion. It is very helpful to improve the quality of this manuscript. We checked almost all the references using in the previous Discussion, and renewed the discussion and also some references partly. Please see the manuscript with track.

  1. In some sentence, English appears not to be adequate.

Response: Thank you very much. We have checked the whole manuscript. We checked the language problem thorough the whole manuscript, and let a native speaker in the USA and one in Switzerland to help us polish the language.

  1. To use a space between the number and the unit, as 20 °C; and not to use a space between number and percentage, as 10%, for example.

Response: Thank you very much. We checked all the °C and %, and corrected them.

  1. Conclusion: please include limitations and future research area!

Response: Thank you very much. We added the implication and future research area in the conclusion part as follows.

This study highlights the significant interactions among soil properties, nematodes and microbial communities, which implies the necessity of simultaneous study on multiple factors when exploring soil function changes as influenced by diversified cover crop patterns in tea gardens.

Further studies should be designed on a multi-year basis, and long-term studies of automatic detection of soil temperature and humidity should be considered into studies on soil biodiversity as affected by various cover crop patterns. In addition, both beneficial and harmful nematode and microorganism species should be analyzed and these interactions should be studied in more details in the future study.

  1. References must be revised.

Response: Thanks. We checked all the references throughout the manuscript

Once again, we appreciate you very much for the valuable suggestions. We hope these revisions could meet with your approval. We would like to revise again, in case they do not meet your demand. We learn a lot from you.  Thank you very much.

Reviewer 2 Report

The authors investigated the effects of different cover crop patterns on soil microorganism and nematode communities in tea garden, they found that the soil nematodes and microbial communities showed significant relationships with soil physical and chemical properties under different cover crop patterns. The results clarify the below-ground biological function of tea garden, and provide important theoretical basis for agricultural management. The topic of the study fits the scope of the journal, and the study has contributed some interesting new data points. However, the language in the manuscript needs clarification and corrections. Writing/clarity need additional attention. The soil physical and chemical properties in the result section should be described in detail.

The following issues should be fixed.

  1. Line 22, line 26 and line 669, “physical and chemical soil properties” should be revised as “soil physical and chemical properities”.
  2. Line 39, “Cui et al., 2018”should be revised as “;Cui et al. 2018”.
  3. Please correct the author namesof reference (Guohua et al. 2017) in line 45, (Xiquan et al. 2005) in line 61, (Yangfang 2010) in line 69,(Yuji et al. 2018) in line 76, (Xiaofang et al. 2004) in line 150, (Xiaodong et al. 2016) in line 196, and the author names in the referencelist.
  4. Line 114 please add “in”beforeMay…..
  5. Line 133, Luebbe but in line 134 Lubbe.
  6. Line 144, “Microbial carbon”should be revised as “Microbial biomass carbon”, “microbial nitrogen”should be revised as “microbial biomass nitrogen”.
  7. Please revise “where”in line 174, 179,182, 201 and 206 as “Where”.
  8. Please revise “bacterial”as “bacteria”in line 196.
  9. Line 322, significantly significant?
  10. Line 583 and 628, please revise “Total”as “total”.
  11. In the title of figure 1, the meaning of A0, A1, A2, and A3 should be given.
  12. Line 608, “of”in the title of figure 2 should be deleted.
  13. Please check the unit of MBC and MBN in table 1.

Author Response

The authors investigated the effects of different cover crop patterns on soil microorganism and nematode communities in tea garden, they found that the soil nematodes and microbial communities showed significant relationships with soil physical and chemical properties under different cover crop patterns. The results clarify the below-ground biological function of tea garden, and provide important theoretical basis for agricultural management. The topic of the study fits the scope of the journal, and the study has contributed some interesting new data points. However, the language in the manuscript needs clarification and corrections. Writing/clarity need additional attention. The soil physical and chemical properties in the result section should be described in detail.

 Dear Reviewer#2,

We appreciate you very much for giving us this valuable opportunity to revise this manuscript, and thank you very much for the helpful suggestions. The whole manuscript was revised carefully according to your suggestions.

We have also asked a native English speaker in the USA and also one in Switzerland to help us polish the language. Plus, the soil physical and chemical properties in the result section has been described in more detail.

Detailed responses are as follows:

The following issues should be fixed.

  1. Line 22, line 26 and line 669, “physical and chemical soil properties” should be revised as “soil physical and chemical properities”

Response: Thank you for your valuable suggestion. We corrected them and checked the whole manuscript.

  1. Line 39, “Cui et al., 2018”should be revised as “;Cui et al. 2018”.

Response: Thank you very much. We revised as you suggested.

  1. Please correct the author namesof reference (Guohua et al. 2017) in line 45, (Xiquan et al. 2005) in line 61, (Yangfang 2010) in line 69,(Yuji et al. 2018) in line 76, (Xiaofang et al. 2004) in line 150, (Xiaodong et al. 2016) in line 196, and the author names in the referencelist.

Response: Thank you very much for your careful comment. We checked all these references, and corrected them. Also, we checked all the reference list.

Yan, G.H., Hou, Y.J., Chen, Gang.,Dai, Y.F., Zhi, Y.K., Sun, Q., Zou, L.M. 2017. The Formation and Loss of Biodiversity. Sichuan forestry exploration and design. 02, 16-24. (In Chinese with English Abstract)

Yao, X.D., Wang, W., Zeng, H., 2016. Application of phospholipid fatty acid method in analyzing soil microbial community composition. Microbiology China. 43, 2086-2095.

Shen, X.Q., Yang, G.P., Liao, M.J. 2005. Isolation of fixed nematode individual genomic DNA and amplification of18S ribosomal RNA gene fragmets. Marine Sciences. 2005, 33-36. (In Chinese with English Abstract)

Shang, Y.F. 2010. Response of soil nematode community to forest vegetation restoration in shimen mountain, dalian. Master's Thesis. Liaoning normal university, China. (In Chinese with English Abstract)

Jiang, Y.J.,  Zhou, Hu., Chen, L.J., Yuan, Y., Fang, H., Luan, Lu., Chen, Y., Wang, X.Y., Liu, M.Q., Li, H.X; Peng, X.H; Sun, B, 2018. Nematodes and microorganisms interactively stimulate soil organic carbon turnover in the macroaggregates. Frontiers in microbiology, 9. DOI: 10.3389/fmicb. 2018.02803.

Yao, X.D., Wang, W., Zeng.H., 2016. Application of phospholipid fatty acid method in analyzing soil microbial community composition. Microbiology, 43(9): 2086-2095. DOI:10.13344/j.microbiol.china.160014.

Zhang, A.L., Zhao, J.N., Liu, H.M., Hong, J., Zhang, N.Q., Yang, D.L. Effects of nitrogen addition on soil nematode community characteristics in Stipa baicalensis steppe.Acta Ecologica Sinica, 2018, 38( 10) : 3616-3627. (In Chinese with English Abstract)

Jiang, Y.J.,  Zhou, Hu., Chen, L.J., Yuan, Y., Fang, H., Luan, Lu., Chen, Y., Wang, X.Y., Liu, M.Q., Li, H.X; Peng, X.H; Sun, B, 2018. Nematodes and microorganisms interactively stimulate soil organic carbon turnover in the macroaggregates. Frontiers in microbiology, 9. DOI: 10.3389/fmicb. 2018.02803.

  1. Line 114 please add “in”beforeMay…..

Response: Thank you very much. We added “in” before May.

  1. Line 133, Luebbe but in line 134 Lubbe.

Response: Thank you very much for your careful comment. We checked it, and it should be the Bran Lubbe AA3, Germany. Thus, Luebbe has been corrected to Lubbe.

  1. Line 144, “Microbial carbon”should be revised as “Microbial biomass carbon”, “microbial nitrogen”should be revised as “microbial biomass nitrogen”.

Response: Thanks. The suggestion has been implemented, and we also checked the whole manuscript.

  1. Please revise “where”in line 174, 179,182, 201 and 206 as “Where”.

Response: Thanks. The suggestion has been implemented.

  1. Please revise “bacterial”as “bacteria”in line 196.

Response: Thanks. The suggestion has been implemented.

  1. Line 322, significantly significant?

Response: Thanks. We feel very sorry for the confusion. We deleted “significant”.

  1. Line 583 and 628, please revise “Total”as “total”.

Response: Thanks. The suggestion has been implemented. Also, we corrected the same problem in Fig.4.

  1. In the title of figure 1, the meaning of A0, A1, A2, and A3 should be given.

Response: Thanks. We feel very sorry for the confusion, and added the meaning of A0, A1, A2, and A3 in the title of Figure1 and Table 1.

Note: A0 represents naturally covered of bare land; A1 represents mixed planting with two-species; A2 represents four-species, andA3 represents eight-species. The same below.

  1. Line 608, “of”in the title of figure 2 should be deleted.

Response: Thanks. We feel very sorry for the error, and deleted “of” in the title of figure 2.

  1. Please check the unit of MBC and MBN in table 1.

Response: Thank you very much for your careful suggestion. It should be mg/kg, and we have corrected it.

Once again, we appreciate you very much for your efforts. We hope these responses could meet with your approval. We would like to revise them, in case they do not meet your demands. Thank you very much!

Reviewer 3 Report

The aim of the study was to explore the dynamics of microorganism and nematode communities, to uncover the relationship between soil physical and chemical factors as well as nematodes and microbes, as affected by planting diversified cover crops in a tea plantation.

The experiment was correctly set up, methodically. Appropriate statistical methods were applied in the analysis of the findings. The findings of the study were presented in tabular (figure) forms and described with accompanying discussion, that was conducted relying on current literature related to the study topic.

Beside the positive qualities of the paper, it does however contain some shortcomings, i.e.,

  1. The work does not contain a research hypothesis. it should be defined no research hypothesis
  2. The non-inclusion, in the text, of a position provided in the References: (cit:Calderon et. al. (line 405), Lin et. al. (408 line) Wei et.al. ( line 449), Liu et. al.(line 462),Longbo at.el. (line465)Yao et. al. ( line 531), Mao et.al (line 533)
  3. The non-inclusion in the Reference – Chen et. al 2018 (line 33, and 42)

The findings bring new, and valuable information, especially those interactions among soil, nematodes  and microbial communities as influenced by diversified cover modes in tea gardens.

In conclusion, I assess the reviewed paper as good and recommend its publication in the Int. J. Environ. Res. Public Health.

Author Response

The aim of the study was to explore the dynamics of microorganism and nematode communities, to uncover the relationship between soil physical and chemical factors as well as nematodes and microbes, as affected by planting diversified cover crops in a tea plantation.

The experiment was correctly set up, methodically. Appropriate statistical methods were applied in the analysis of the findings. The findings of the study were presented in tabular (figure) forms and described with accompanying discussion, that was conducted relying on current literature related to the study topic.

Dear Reviewer#3,

We appreciate you very much for the helpful suggestions. We tried our best to revise this manuscript following yours comments. We believe that the entire manuscript has been improved a lot after extensive revisions.

Detailed responses are as follows:

Beside the positive qualities of the paper, it does however contain some shortcomings, i.e.,

  1. The work does not contain a research hypothesis. it should be defined no research hypothesis

Response: Thank you for your valuable suggestion. We added the research hypothesis.

We hypothesized that: (1) diversified cover crop patterns exerted different effects on microorganism and nematode communities in a tea plantation, (2) soil physical and chemical factors are interacted with dynamics of soil nematodes and microbes, as affected by planting diversified cover crops, (3) soil microorganism and nematode communities show different changing trends under the diversified cover crop patterns, depending on seasonal changes.

  1. The non-inclusion, in the text, of a position provided in the References: (cit:Calderon et. al. (line 405), Lin et. al. (408 line) Wei et.al. ( line 449), Liu et. al.(line 462),Longbo at.el. (line465)Yao et. al. ( line 531), Mao et.al (line 533)

Response: Thank you very much for your careful suggestions. We checked all the references throughout the whole manuscript. We deleted these non-inclusion references and also checked all the other references.

  1. The non-inclusion in the Reference – Chen et. al 2018 (line 33, and 42)

Response: Thank you very much for your careful suggestions. We deleted it. Please see the updated references. We checked all the other references.

The findings bring new, and valuable information, especially those interactions among soil, nematodes  and microbial communities as influenced by diversified cover modes in tea gardens.

In conclusion, I assess the reviewed paper as good and recommend its publication in the Int. J. Environ. Res. Public Health.

Once again, on behalf of all co-authors, we really appreciate your patience and enthusiastic participation on our manuscript.

Please see the attachment of the ms with track

Round 2

Reviewer 2 Report

The authors have improved the manuscript and have responded appropriately to my concerns. However, the following issues should be fixed before publishing.

In the title of the paper, I suggest add “soil” before microorganism and nematode communities ...

Line 49, 54, 86, add “,” after et al.

Line 136, 1. Materials and Methods, 1. should be revised as 2.

Line 184, Biomass should be revised as biomass.

Line 185, Microbial Biomass should be revised as microbial biomass.

Line 294, please revise moths as months.

Line 444, insoil should be revised as in soil.

Line 795, table S2, and table S4, bacterial should be revised as bacteria.

In table S5, Totol should be revised as Total.